



# A global streamflow indices time series dataset for large-sample hydrological analyses on streamflow regime (until 2021)

Xinyu Chen[1], Liguang Jiang[1], Yuning Luo[2], and Junguo Liu[1, 3]

[1] School of Environmental Science and Engineering, Southern University of Science and Technology, Shenzhen, 518055, China

[2] State Key Laboratory of Hydrology-Water Resources and Hydraulic Engineering, and College of Hydrology and Water Resources, Hohai University, Nanjing, 210098, China

[3] Henan Provincial Key Lab of Hydrosphere and Watershed Water Security, North China University of Water Resources and Electric Power, Zhengzhou, 450046, China

*Correspondence*: Liguang Jiang (jianglg@sustech.edu.cn)

**Abstract.** With the booming big data techniques, large-sample hydrological analysis on streamflow regime is becoming feasible, which could derive robust conclusions on hydrological processes from a big-picture perspective. However, there is not a comprehensive global large-sample dataset for components of the streamflow regime yet. This paper presents a new time series dataset on global streamflow indices calculated from daily streamflow records after data quality control. The dataset contains 79 indices over seven major components of streamflow regime (*i.e.,* magnitude, frequency, duration, changing rate, timing, variability, and recession) of 5548 river reaches globally. The indices time series in the dataset are available until 2021, the lengths of which vary from 30 to 215 years with an average of around 66 years. Restricted-access streamflow data of typical river basins in China are included in the dataset. Compared to existing global datasets, this global dataset covers more indices, especially those characterizing the frequency, duration, changing rate, and recession of streamflow regime. With the dataset, research on streamflow regime will become easier without spending time handling raw streamflow records. This comprehensive dataset will be a valuable resource to the hydrology community to facilitate a wide range of studies, such as studies of hydrological behaviour of a catchment, streamflow regime prediction in data-scarce regions, as well as variations in streamflow regime from a global perspective.

## 1 Introduction

Streamflow regime plays a vital role not only in human life and activities but also in the native biodiversity, ecosystem integrity, and biogeochemical cycles (Poff et al., 1997; Paine, 2019; Palmer and Ruhi, 2019). Because of the effects of anthropogenic activities and climate change especially in the last decades, streamflow regimes of many rivers worldwide have been changing, threatening the water security (Torabi Haghighi et al., 2021; Tonkin et al., 2018). A large number of studies have been done to reveal the streamflow regime shifts, their causes and consequences (Worku et al., 2014; Brouziyne et al., 2021; Sauquet et al., 2021; Lane and Kay, 2021; Yin et al., 2018). Palmer and Ruhi (2019) found that the dam building, diversion or abstraction of water, clearing of land, and climate change increasingly degraded the river ecosystems by altering their streamflow regimes. Barichivich et al. (2018) indicated that the streamflow regime shifts over the Amazon basin in magnitude and frequency, which has caused major human suffering and disturbance to the rainforest ecosystems, are driven by strengthened Walker circulation.

In order to analyse the streamflow regime shifts, the critical components of the streamflow regime, *i.e.*, magnitude, frequency, duration, timing, and rate of change, were proposed to characterize the entire range of streamflow regime and specific hydrologic phenomena (Poff and Ward, 1989; Poff et al., 1997; Richter et al., 1996). By using these components, features of streamflow regime can be considered explicitly, and therefore these components are widely used (Olden and Poff, 2003; Worku et al., 2014; Palmer and Ruhi, 2019; Shih et al., 2022; Jacobson et al., 2022; Harmon et al., 2022; Wasko et al., 2020). Besides,





inspired by this, more and more indices, signatures, and components are proposed to represent different aspects of streamflow regime (Clausen and Biggs, 2000; Baker et al., 2004; Clark et al., 2009; Botter et al., 2013; Mcmillan et al., 2017). For example, Baker et al. (2004) presented a new flashiness index based on daily streamflow to characterize the flashiness of streamflow regime, which was later widely used (Gnann et al., 2021b). However, except for several basic indices of magnitude and frequency like the annual maximum streamflow (Do et al., 2017; Barichivich et al., 2018), there are few large-sample and

global studies on other components such as timing, variability, and rate of change. Gudmundsson et al. (2018) found that there was no any study analysing time series of the variability (e.g., standard deviation, coefficient of variation, Gini coefficient, and the inter quartile range) and timing (e.g., the timing of annual minimum flow, day of minimum 7-day mean streamflow, and day of maximum 7-day mean streamflow) of daily streamflow on a global scale.

Large-sample hydrology is a way to go beyond individual case studies and to draw robust conclusions on hydrological

processes from a big-picture perspective (Gupta et al., 2014; Addor et al., 2020). Currently, due to the increasing availability of large-sample hydrology datasets, as well as the booming big data techniques, more and more large-sample hydrological studies have been appearing, significantly advancing the hydrology science (Sun et al., 2021; Troin et al., 2022; Lane et al., 2022; Goeking and Tarboton, 2022; Nearing et al., 2021; Gnann et al., 2021a; Gudmundsson et al., 2021). To perform large-sample hydrological analysis, large-sample hydrological datasets based on gauged flow data are mostly needed. Addor et al.

(2017) presented the CAMELS (Catchment Attributes and MEteorology for Large-sample Studies) dataset, which synthesized various datasets (including meteorological forcing and gauged daily streamflow time series) to describe attributes of catchments and catchment behaviours in the contiguous United States. Afterwards, diverse versions of CAMELS or CAMELS-like datasets were presented for different countries, such as the Great Britain (Coxon et al., 2020), Chile (Alvarez-Garreton et al., 2018), Brazil (Chagas et al., 2020), Australia (Fowler et al., 2021), Central Europe (Klingler et al., 2021), France (Delaigue

et al., 2022), and Germany (Ebeling et al., 2022). Besides, there are also regional streamflow indices datasets (Tramblay et al., 2021) and global streamflow indices and metadata archive (GSIM) for analyses on regional and global streamflow characteristics (Do et al., 2018; Gudmundsson et al., 2018). Unfortunately, many original records of gauged streamflow are not open access, and are forbidden to be shared due to certain policies. In contrast, the data products derived from restricted-access streamflow records are usually allowed to be shared. Thus, streamflow indices datasets not only facilitate the research

on streamflow regimes, but also are good alternatives when the original records of streamflow are hard to access (Tramblay et al., 2021; Mcmillan et al., 2017).

GSIM covers time-series indices of more than 30000 stations worldwide, which represents the water balance, the seasonal cycle, low flows, and floods, with the latest streamflow data until 2017. It is without doubt one of the most popular datasets which facilitate large-sample research on global streamflow. However, GSIM only includes the streamflow regime components

that characterize the magnitude, timing, and variability, but does not include components characterizing the frequency, duration, changing rate, and recession of streamflow regime. In fact, these components are very useful to fully characterize the flow regime, understand its functions, and analyse its variations. For instance, the frequency and duration of streamflow regime are very important in describing various flow events. Gehrke et al. (1995) found that in the Murray-Darling river system, the altered frequencies of high and low flow events have a significant impact on the species diversity of fish communities. Colls

et al. (2019) analysed the frequency and duration of zero flow events over 33 Mediterranean streams in NE Iberian Peninsula and indicted that longer duration of zero flow events significantly decreases gross primary production promoting heterotrophy. Changing rate is an important factor affecting the lives of aquatic species. Cushman (1985) reviewed the changing rate of streamflow regime below hydroelectric facilities and found that the rapid changes of river stage has damaged aquatic species by wash-out and stranding. Palmer and Ruhi (2019) added that the increase of changing rate during storms results in elevated

concentrations of pollutants, which is also harmful to the lives of aquatic species. The recession of streamflow reflects the low-flow behaviour of a catchment and plays a vital role in both flow-biota-ecosystem processes nexus and water management.





Boggaart et al. (2016) used streamflow recession patterns to unravel the role of climate and humans in landscape co-evolution. Rood et al. (1995) did a study on the recession of streamflow and presented that the accelerated flood recession resulted in the failure of seedling establishment and the decline of riparian cottonwoods along the St. Mary River. Cheng et al. (2021)

indicated that analysis on recession of flood is critical for flood risk reduction and water use in the Huaihe River Basin. In this regard, a more comprehensive indices dataset than GSIM is needed. Actually, Tramblay et al. (2021) presented the African Database of Hydrometric Indices (ADHI, 1950–2018) with a more comprehensive streamflow indices, but it is geographically limited to the Africa. There is no comprehensive global large-sample dataset of different components of streamflow regime, which hinders research on streamflow regime, especially on a global scale.

In this paper, we augmented the gauged daily streamflow data from Global Runoff Data Centre (GRDC) by collecting streamflow data from India-Water Resources Information System (WRIS), Arctic Great Rivers Observatory (ArcticGRO), and China Hydrological Yearbooks (CHY) to build a daily streamflow data collection. After that, quality control was done to guarantee reasonable values as well as a longer record length. Next, indices spanning 7 components of streamflow regime, *i.e.*, magnitude, frequency, duration, changing rate, timing, variability, and recession, were defined and calculated to build a new

global streamflow indices dataset. Finally, an exemplary analysis on the temporal concentration of streamflow on a global scale was presented to illustrate the use of the dataset.

## 2 Data sources and processing

### 2.1 Data collecting

The daily streamflow data of the daily streamflow data collection are from Global Runoff Data Centre (GRDC) at

https://www.bafg.de/GRDC/EN/02_srvcs/21_tmsrs/riverdischarge_node.html and other three sources, i.e., India-Water Resources Information System (WRIS) at https://indiawris.gov.in/wris/#/, Arctic Great Rivers Observatory (ArcticGRO) at https://arcticgreatrivers.org/, and China Hydrological Yearbooks (CHY). These data sources are all publicly available except the CHY. The original records of streamflow in CHY are restricted-access and hard to collect, and thus only some streamflow data of typical river basins were collected. The total amount of hydrological stations in the data collection is 15408, the

summary of these stations is shown in Fig. 1. It should be noted that 9171 stations have daily streamflow records, while the others only have basic information of stations without daily streamflow records. The daily streamflow record lengths vary from 1 to 215 years.

The GRDC is very comprehensive in terms of the record length and spatial coverage of gauged daily streamflow data. The spatial density and record lengths of streamflow time series in the Americas, Europe, Southern Africa, Western Africa, and

Oceania are high and long (Fig.1a). There are 10711 hydrological stations in GRDC, out of which 8552 stations have daily streamflow records. The daily streamflow records range from 1806 to 2021 and the lengths vary between 1 and 215 years. However, although there are enormous stations in Asia included in GRDC, almost all of their record lengths are shorter than 10 years, which largely reduces their value for hydrological analyses. Therefore, we combined GRDC with ArcticGRO, CHY, and WRIS to build an augmented data collection. India-WRIS provides water resources data and information of watersheds in

India for planning, development, and integrated water resources management. It includes 4648 stations, but only 570 stations have daily streamflow records. The daily streamflow records ranging from 1960 to 2021 are used to build the data collection. ArcticGRO provides essential data about the biogeochemistry and discharge of the largest Arctic rivers. It covers 16 stations with daily streamflow records ranging from 1927 to 2021 and the record lengths vary from 5 to 95 years. 33 stations from CHY with daily streamflow records ranging from 1947 to 2020 are used to build the data collection. The record lengths vary

from 1 to 73 years. These stations are of typical river basins and located in 7 largest river basins in China, *i.e.*, Yangtze River

Basin, Yellow River Basin, Huai River Basin, Haihe River Basin, Songhua River Basin, Liao River Basin, and Pearl River Basin.

In our streamflow data collection, streamflow record lengths at more than 700 stations are more than 100 years, most of which are distributed in the United States and Europe. About 6000 stations (67%) are of lengths between 30-100 years, while the others (33%) are of lengths less than 30 years (Fig.1b). As for the temporal distribution of numbers of stations with gauged daily streamflow data in different years, the number increases from 1900 to 1987 at a peak of around 7500, and it drops from 1987 to 2021 at a bottom of nearly 1400 (Fig.1c).

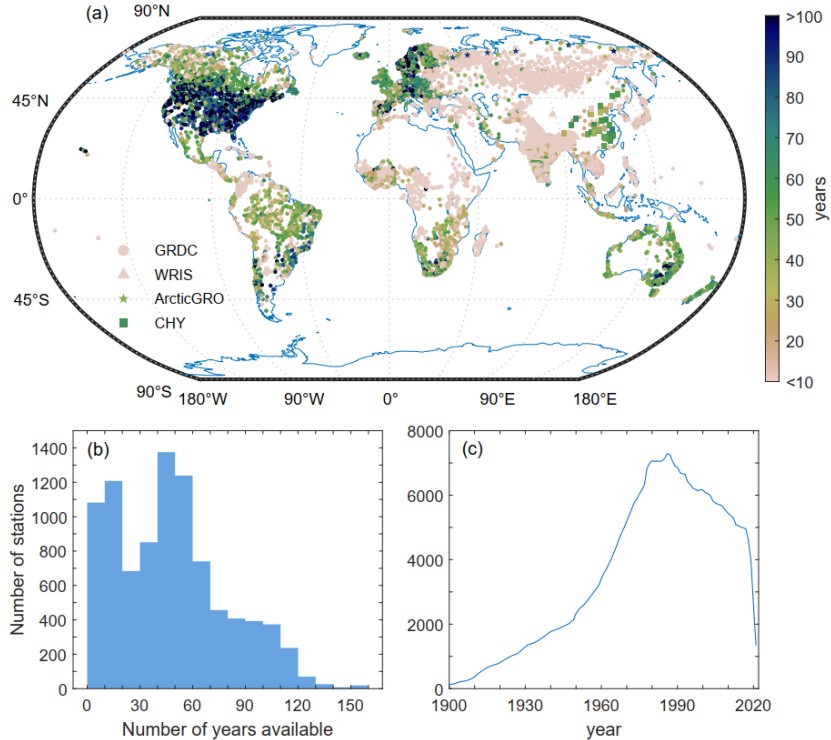

**Figure 1. A summary of the data collection. (a) shows the spatial distribution, record lengths, and sources of daily flow time series. (b) illustrates number of stations available for different record lengths, and stations without gauged streamflow data are not included. (c) shows number of stations with gauged streamflow data available every year from 1900 to 2021.**

### 2.2 Data quality control

In order to build a high-quality, long time series, and reliable global streamflow indices dataset, data quality control on the data collection is needed before further calculations. An Automatic detection method was applied to identify the suspect observations according to three quality control criteria detailed below, and modifications were done to the suspect observations:

1. Considering possible mistakes made by instruments and humans, negative daily streamflow values may occur in the data collection, which is non-physical and must be revised. If a daily streamflow value is a negative number, this value will be regarded as a suspect value, and then this value will be changed into an average value of the two adjacent values. A negative value detected in the time series of Jenapur at Brahmani, India from WRIS is shown for the purpose of illustration (Fig.2a).



2.  If there are more than 10 consecutive equal values bigger than 0 and 50th percentile of daily streamflow in corresponding year, they will be regarded as suspect values, and then they will be changed into missing values. There are many reasons for the consecutive equal values. It may occur because of instrument failure, *i.e.*, damaged sensors and ice jams, or flow regulations (Gudmundsson et al., 2018). Moreover, it also happens when the day-to-day fluctuations of streamflow are below the sensitivity of the employed sensor (Gudmundsson and Seneviratne, 2016). Because this issue usually occurs during the low flow period, we choose the 50th percentile of daily streamflow as the threshold to exclude those abnormal cases. The selection of 10 days is according to Gudmundsson et al. (2018). Consecutive equal values found in the time series of Etemba at Omaruru, Namibia from GRDC are shown for the purpose of illustration (Fig.2b).

3.  According to Gudmundsson et al. (2018), if $\log(Q+0.01)$ is bigger or smaller than the mean value of $\log(Q+0.01)$ plus or minus 6 times the standard deviation of $\log(Q+0.01)$, where $Q$ is a daily streamflow value and the mean value and standard deviation are calculated in a 5-day window centred on the calendar day of $Q$, the $Q$ will be regarded as an outlier and changed into an average value of the two adjacent values. Following Gudmundsson et al. (2018), the 6 standard-deviation threshold is reasonable, because it keeps a balance between screening out outliers that could come from instrument malfunction and retaining the extreme floods or low flows. An outlier detected in the time series of Luanxian at Luanhe River, China from CHY is presented for the purpose of illustration (Fig.2c).

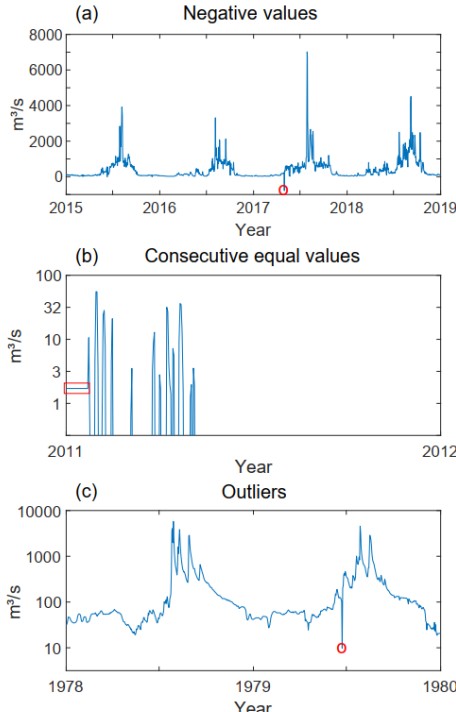

**Figure 2. Three example time series illustrating issues detected by the three quality control criteria (highlighted in red). (a) shows the negative value detected in the time series of Jenapur at Brahmani, India from WRIS. (b) presents the consecutive equal values found in the time series of Etemba (64733002) at Omaruru, Namibia from GRDC. (c) illustrates one outlier in the time series of Luanxian at Luanhe River, China from CHY. The common logarithmic axis is used in (b) and (c).**

After automatic detection and modification of suspect observations, a final data quality control is applied. That is, the time series with the date of latest records before 2000 or lengths shorter than 30 years were removed, since they are outdated and less convincing for streamflow regime analyses. In the end, 5548 time series were remained for the calculation of indices.



165    The availability of daily streamflow data after data quality control is shown in Fig.3. Nearly 1500 stations have a record length
       of 72 years from 1950 to 2021 with only 5% missing data, and around 3000 stations have a record length of 52 years from
       1970 to 2021 with 10% missing data (Fig.3a). At approximate 800 stations, their streamflow records are not available since
       2013 (Fig.3b). By comparison, at more than 4000 stations, streamflow records are available even after 2017. At around 2400
       stations, the latest streamflow records are available till 2020 or 2021.

Fig.3c shows the number of stations available every year from 1900 to 2021 with various missing data rates. All curves in
       Fig.3c show similar trends. The number of stations gradually rises from 1900 to its peak at around 1985, and then it keeps
       slightly dropping until around 2010, followed by a drastic decrease until 2021. With regard to the missing data rate, the stations
       numbers are around 4500 with missing data rate < 1% and around 4700 with missing data rate < 5% in 1980-2005. In 2020,
       there are still around 1200 stations available with missing data rate < 1%.


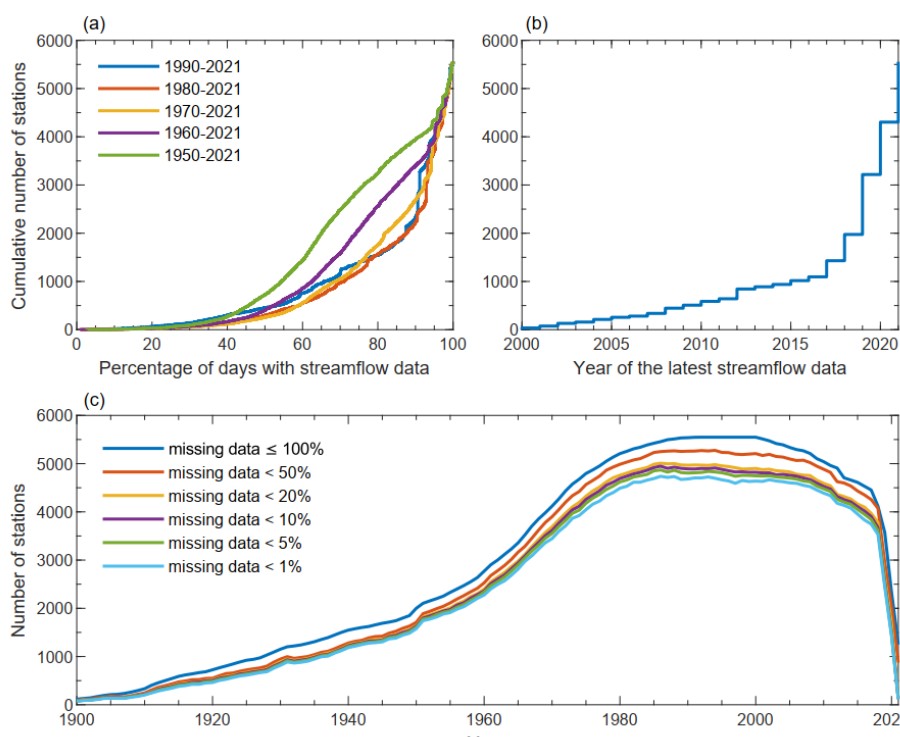

**Figure 3. Availability of daily streamflow data after data quality control. (a) and (b) show the cumulative distribution of stations
corresponding percentage of days with streamflow data in different record lengths and year of the latest streamflow data,
respectively. (c) presents number of stations available every year from 1900 to 2021 with various missing data rates.**


## 3  Streamflow indices

### 3.1  Indices definition and calculation

       Table 1 describes 79 streamflow time-series indices that characterize seven components of streamflow regime, *i.e.*, magnitude,
       frequency, duration, changing rate, timing, variability, and recession. These indices were calculated from the streamflow data
collection after the data quality control, and most of them are computed with the Toolbox for Streamflow Signatures in



Hydrology (TOSSH, available at the address: https://github.com/TOSSHtoolbox/TOSSH) (Gnann et al., 2021b). Only data in years with less than 5% missing data are included in indices calculation.

The magnitude of streamflow regime can reflect the amount of streamflow from various perspectives. The corresponding indices include: (i) maximums of consecutive 1, 3, 7, and 30 day streamflow averages and their percentages, which indicate
the magnitude and concentration of high flows and floods; (ii) minimums of consecutive 1, 7, and 30 day streamflow averages, which indicate the magnitude of low flows; (iii) various percentiles of streamflow; (iv) monthly/annual mean flow, which is usually used for water resources analysis; (v) high/low flow event threshold (Clausen and Biggs, 2000; Olden and Poff, 2003); (vi) runoff/baseflow magnitude (Horner, 2020), which indicates the magnitude of difference between the maximum and the minimum of runoff/baseflow.

The frequency of streamflow regime is how often a flow of specific magnitude recurs over some specified time intervals (Poff et al., 1997). The corresponding indices include the ratios of days with streamflow reaching specific thresholds to the total days and the numbers of streamflow events (floods, high flows, low flows and so on) with various thresholds. The duration is the period of time during which a streamflow event lasts. Annual mean durations of streamflow events are calculated as indices.

The changing rate, or flashiness, means how fast and frequently streamflow alters from one magnitude to another (Poff et al.,
1997; Baker et al., 2004). A flashy river basin has a very quick and sensitive response to incoming water like precipitation, and its streamflow rises and falls very rapidly. The Richards-Baker flashiness index (Baker et al., 2004), and the mean and median of all positive/negative differences between consecutive daily streamflow values (The Nature Conservancy, 2009) are used to quantify the flashiness of streamflow. Rising limb density is an index that describes the flashiness of the catchment response; for example, a small value means a smooth hydrograph (Sawicz et al., 2011).

The timing of streamflow regime is the temporal distribution of streamflow in a year (Court, 1962), which is charactered by the start date of flood season, half flow date, half flow interval, momentary maximum date, and minimum consecutive 7 day flow date in the indices dataset. To calculate the half flow date and half flow interval, the start of the water year is needed (Court, 1962). Although it is widely used that the start of the water year is 1 October in the Northern Hemisphere and 1 July in the Southern Hemisphere, the actual starts of the water year vary greatly even in different river basins of one hemisphere
because of different geographical features, climates and so on. In the indices dataset, we use the start date of flood season as the start of the water year. The start date of the flood season for a specific station is the median of start dates of consecutive 180 days of which the streamflow average is the biggest in one calendar year.

The indices of variability characterize the variability of streamflow regime from different perspectives (Gudmundsson et al., 2018). (i) Variance of streamflow time series provides information on the total variability of streamflow. (ii) Coefficient of
variation of streamflow provides a relative measure of variability that is independent of the mean flow. (iii) Quartile-based coefficient of variation of streamflow time series provides information about the width of the distribution centre and is less sensitive to outliers. (iv) Ratio of the maximum to median of streamflow quantifies the deviation of maximum. (v) The Gini coefficient is an index to measure the inequality among values of flow duration curve (Gudmundsson et al., 2018). (vi) Slope of flow duration curve is an index of the variability of the seasonal water balance, which shows the difference between high
and low flows (Mcmillan et al., 2017). Besides, it is also sensitive to vertical redistribution of soil water between quick flow and slow flow. (vii) Slope of distribution of peaks is an index for measuring the differences between peak discharges (Euser et al., 2013). (viii) Variability index was a measure for variability among values of flow duration curve (Lane and Lei, 1950). A river with higher variability index tend to have higher percentage of surface runoff and lower storage (Estrany et al., 2010).

Recession is a component of streamflow regime which characterizes the recession of streamflow. The smoothed minima
baseflow separation method of the UK Institute of Hydrology (UKIH) (1980) is used for baseflow separation required in the calculation of recession indices. Generally, a river with low baseflow index value has a great number of floods and low flows,





and its streamflow regime is highly variable (Singh et al., 2019). Baseflow index has been commonly used in regional low flow studies, impacts of climate change on groundwater resources, and flood responses of river basins to storm events. Baseflow recession constant is a proxy for drainage efficiency of baseflow after being recharged, which is related to the

watershed hydraulic conductivity, soil porosity, and hydraulic gradient (Safeeq et al., 2013). According to Safeeq et al. (2013), a river basin with high baseflow recession constant has a shallow subsurface flow-dominated fast draining system, whereas a river basin with low baseflow recession constant has a groundwater-dominated slow draining system.

**Table 1. Streamflow indices for seven components of the streamflow regime. Index name means the variable name used in the indices dataset. There are two temporal resolutions. Y (yearly) means one value for one year of the time series, and MY (multi-year) means**

**one value for the whole time series.**

| Category | Index name | Units | Resolution | Definition |
|---|---|---|---|---|
| Magnitude | *Qmax1, Qmax3, Qmax7, Qmax30* | $m^3/s$ | Y, MY | Maximums of consecutive 1, 3, 7, and 30 days streamflow averages. For example, *Qmax7* means the maximum of consecutive 7-day streamflow averages (Olden and Poff, 2003). |
| | *Qmax1p, Qmax3p, Qmax7p, Qmax30p* | - | Y | The percentages of maximums of consecutive 1, 3, 7, and 30 days streamflow accumulation amount, which are the maximums divided by the total of annual streamflow. |
| | *Qmin1, Qmin7, Qmin30* | $m^3/s$ | Y, MY | Minimums of consecutive 1, 7, and 30 days streamflow averages (Olden and Poff, 2003). |
| | *Q1st, Q5th, Q10th, Q25th, Q50th, Q75th, Q90th, Q95th, Q99th* | $m^3/s$ | Y, MY | The 1st, 5th, 10th, 25th, 50th 75th, 90th, 95th, and 99th percentiles of daily streamflow (The Nature Conservancy, 2009; Olden and Poff, 2003). For example, *Q50th* means the median of streamflow time series. |
| | *Qmean1, Qmean2, Qmean3, Qmean4, Qmean5, Qmean6, Qmean7, Qmean8, Qmean9, Qmean10, Qmean11, Qmean12, Qmean* | $m^3/s$ | Y, MY | Monthly and annual mean flows. For example, *Qmean6* means the monthly mean flow of June; *Qmean* is the annual mean flow. |
| | *Qhigh, Qlow* | $m^3/s$ | MY | High and low flow event thresholds. *QHigh* equals 9 times *Q50th* (Clausen and Biggs, 2000); *Qlow* equals 0.2 times *Qmean* (Olden and Poff, 2003). |
| | *RM, BM* | $m^3/s$ | Y | Runoff magnitude and baseflow magnitude. *RM* and *BM* are the differences between the maximum and minimum of streamflow and baseflow respectively (Horner, 2020). |
| Frequency | *FreH, FreL, FreZ* | - | Y | Frequencies of high flow (*FreH*), low flow (*FreL*), and zero flow (*FreZ*) days. *FreH* is the ratio of days with streamflow bigger than *Qhigh* to the total days; *FreL* is the ratio of days with streamflow less than *Qlow* to the total days; *FreZ* is the ratio of days with zero streamflow to the total days (Addor et al., 2018). |
| | *Fre1st, Fre5th, Fre95th, Fre99th* | - | Y | Frequencies of days with streamflow bigger than or smaller than thresholds of the 1st, 5th, 95th, and 99th streamflow percentiles. *Fre1st /Fre5th* is the ratio of days in one year with streamflow less than the 1st/5th percentile of the whole multiyear streamflow time series to the days of one year; *Fre95th /Fre99th* is the ratio of days in one year with streamflow bigger than the 95th/99th percentile of the whole multiyear streamflow time series to the days of one year. |
| | *NumH, NumL, NumZ* | - | Y | Numbers of streamflow events with thresholds of *Qhigh*, *Qlow*, and zero (Olden and Poff, 2003). |
| | *Num1st, Num5th, Num95th, Num99th* | - | Y | Numbers of streamflow events with thresholds of the 1st, 5th, 95th, and the 99th percentile of the whole multiyear streamflow time series (Olden and Poff, 2003). |
| Duration | *DurH, DurL, DurZ,* | days | Y | Mean duration of streamflow events with thresholds of *Qhigh*, *Qlow*, and zero (Westerberg and Mcmillan, 2015). |
| | *Dur1st, Dur5th, Dur95th, Dur99th* | days | Y | Mean duration of streamflow events with thresholds of the 1st, 5th, 95th, and 99th percentiles of the whole multiyear streamflow time series. |
| Changing rate | *RBFI* | - | Y, MY | Richards-Baker flashiness index (Baker et al., 2004). |
| | *RLD* | - | Y, MY | Rising limb density (*RLD*) is a ratio of the number of rising limbs to the number of rising hydrograph (Sawicz et al., 2011). |
| | *RRmean, RRmedian, FRmean, FRmedian* | $m^3/s$ | Y | *RRmean* and *RRmedian* are the mean and median of all positive differences between consecutive daily streamflow values in a year; *FRmean* and *FRmedian* are the mean and median of all negative |





| Category | Index name | Units | Resolution | Definition |
|---|---|---|---|---|
| | | | | differences between consecutive daily streamflow values in a year (The Nature Conservancy, 2009). |
| Timing | FSS | days since 1 January | Y, MY | FSS is the start date of flood season, which is defined as the start date of the consecutive 180 days whose streamflow average is the biggest in specific calendar year. It is calculated as the following: calculate a sliding average streamflow time series by applying sliding average method to the whole streamflow time series with a sliding window of 180 days; found the maximums of every calendar year in the averaged streamflow time series; start dates of corresponding sliding windows are FSSs of every calendar year. |
| | HFD | days | Y, MY | Half flow date (HFD) is the date on which half of a water year's total streamflow has passed since start of the water year (Court, 1962). |
| | HFI | days | Y, MY | Half flow interval (HFI) is the time span between the date on which a quarter of a water year's total streamflow has passed since start of the water year and the date on which three quarters of a water year's total streamflow has passed since start of the water year (Court, 1962). |
| | MMD | days since 1 January | Y, MY | Momentary maximum date (MMD) is the date when the maximum streamflow occurs (Court, 1962). |
| | MC7FD | days since 1 January | Y, MY | Minimum consecutive 7 day flow date (MC7FD) is the date when the minimum of consecutive 7 day flow average occurs (Gudmundsson et al., 2018). |
| Variability | VY | - | Y, MY | Variance of streamflow time series (Clausen and Biggs, 2000). |
| | COVY | - | Y, MY | Coefficient of variation of streamflow time series (Clausen and Biggs, 2000). |
| | QCV | - | Y, MY | QCV means quartile-based coefficient of variation of streamflow time series, which is calculated as (Q75th-Q25th)/Q50th. |
| | RMM | - | Y, MY | Ratio of Qmax1 to Q50th. |
| | GNC | - | Y, MY | Gini coefficient (Gudmundsson et al., 2018). |
| | SFDC | - | Y, MY | Slope of flow duration curve (SFDC) is the slope of flow duration curve between 33rd and 66th percentiles of streamflow (Mcmillan et al., 2017). |
| | SDP | - | MY | SDP is the slope of distribution of peaks, which is the slope between the 10th and 50th of a flow duration curve constructed by only considering hydrograph peaks (Euser et al., 2013). |
| | VI | - | Y, MY | Variability index (VI) is the standard deviation of the common logarithms of streamflow determined at 10% intervals from 10% to 90% of the flow duration curve (Lane and Lei, 1950; Estrany et al., 2010). |
| Recession | BFI | - | Y, MY | Baseflow index (BFI) is the ratio of baseflow volume to streamflow volume over a specific time period (Singh et al., 2019). |
| | BRC | - | Y, MY | BRC is baseflow recession constant. Hydrograph recession assuming exponential recession behaviour is given by $Q_t = Q_0 e^{-kt}$, where $Q_t$ is the streamflow at time $t$ (day), $Q_0$ is the streamflow at the beginning of the recession, and $k$ is the BRC (Safeeq et al., 2013). The master recession curve, which combines individual recession segments, is constructed by using the adapted matching strip method and then used for the calculation of BRC (Posavec et al., 2006). |

Note: For most indices, the calculation on multi-year scale is using the same algorithm as the calculation on yearly scale except that the used time series is the whole multi-year time series rather than one year segment. For indices including *FSS, HFD, HFI, MMD*, and *MC7FD*, the multi-year values are the medians of yearly values.

**3.2 Example streamflow indices time series**

To give a first impression of streamflow indices time series, Fig. 4 shows some example streamflow indices time series for the seven components of the streamflow regime of Nashwaak River at Durham Bridge, Canada at yearly resolution. It is obvious that the *Qmax1* is increasing as well as the *Qmean* while *Qmin7* has no obvious trend, which accompanies an upward trend of *RM*. These trends indicate that the magnitude of high flow is increasing. Moreover, the *Num99th* and *Dur99th* are also

increasing, which means the number and lasting time of flood are rising. To make matters worse, the *RBFI* and *RRmean* are

obviously climbing, too. In contrast, the *FRmean* is decreasing. It means the streamflow regime of Nashwaak River is becoming more and more flashy with a higher rising-dropping speed of floods. Besides, the *BFI* also shows a downward trend, which indicates worse flow regulations of the river basin. In conclusion, these shifts of the streamflow regime components show that the floods have grown in intensity and flood forecasting and protection are becoming more important.


**Figure 4. Example streamflow indices time series for seven components of the streamflow regime of Nashwaak River at Durham Bridge, Canada at yearly resolution. Please refer to Table 1 for the names and units of indices.**



## 4 An exemplary application

An example investigation into the trend and abrupt change of temporal concentration of streamflow on a global scale has been
done to illustrate a use of the streamflow indices time series dataset. Fig. 5 shows the trends and abrupt change points of half-
flow interval (HFI) time series on a global scale with a 95% confidence level. The trends were tested using the Mann-Kendall
trend test (Kendall, 1948) and calculated using the Sen's slope method (Sen, 1968).The abrupt change points were detected by
combining the results of Pettitt test (Pettitt, 1979) and the heuristic segmentation algorithm (Bernaola-Galván et al., 2001) after
detrending. The HFI is an index that quantifies the temporal concentration of streamflow. A small HFI means a half of yearly
discharge is generated in a short time, and thus reflects a concentrated streamflow regime. It is obvious that near and in the
Arctic, almost all rivers' HFIs show significant upward trends, which means streamflow of those rivers has been less
temporally concentrated (Fig. 5a). Feng et al. (2021) found that the zero flow days of Arctic rivers are declining strongly, while
the yearly discharge accelerations altered very slightly, which implies that the streamflow of Arctic rivers is becoming more
evenly distributed within a water year. The causes are likely the reservoir operations and the earlier snowmelt because of the
climate change (Tan et al., 2011; Suzuki et al., 2020; Adam et al., 2007).

It is also noticeable that in the northern part of Australia, the HFIs tend to have an upward trend, while in the southern part,
the HFIs tend to have a downward trend. Actually, there is a downward trend in annual total rainfall in southern part of
Australia and a upward trend in northern part of Australia, and the trend pattern of annual total streamflow is the same as
annual total rainfall (Zhang et al., 2014; Zhang et al., 2016). The trends of HFIs in Australia are probably due to the change of
precipitation, but the cause of the change of precipitation is still under discussion (Dey et al., 2019). Besides, there are clusters
of significant trends of HFIs in the US, middle part of South America, South Africa, Europe, China, and India. There are also
clusters of significant abrupt change points of HFIs as shown in Fig. 5b. In South America, a cluster of significant abrupt
change points occurs in 2010 to 2019, which may be attributed to the strong tropical Atlantic warming and tropical Pacific
cooling as well as the deforestation (Barichivich et al., 2018). In Europe, most of the significant abrupt change points are
before 1970. indicated that strong shifts in the streamflow patterns on a continental scale have occurred in Europe after around
1960, which agrees with our finding. Actually, Europe has experienced pronounced changes inclimate (Fontrodona Bach et
al., 2018) and land cover (urbanization, reforestation, and afforestation; see Fuchs et al. (2013)) since the 1960s, which
probably caused the abrupt change points.



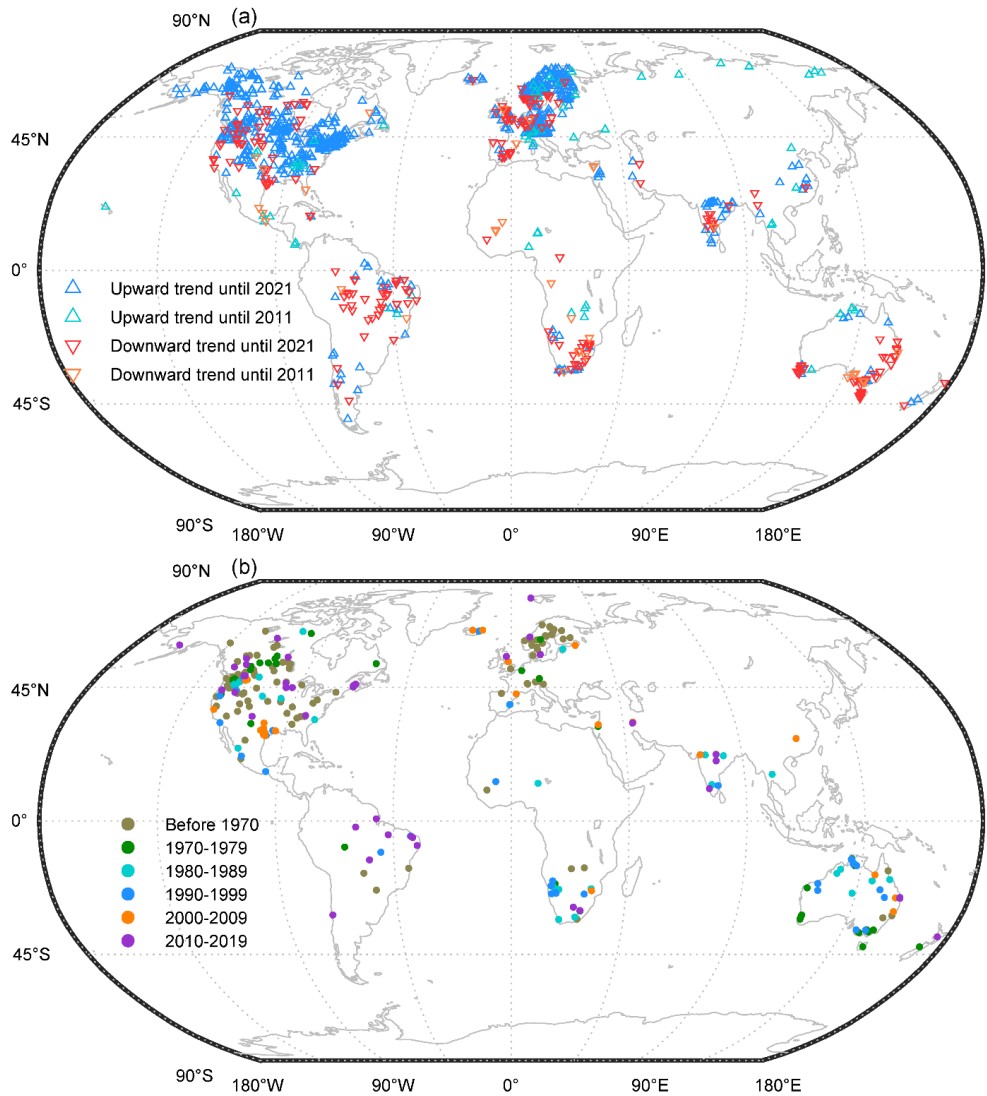

**Figure 5. The (a) trends and (b) abrupt change points of half-flow interval time series on a global scale. All data shown have passed the significance test with a 95% confidence level.**

## 5 Data availability

The global streamflow indices time series dataset is available for download at https://www.scidb.cn/en/s/M32eEb (Chen et al. 2023). There are two folders corresponding to two different data storage ways. One is "MAT" for the files with ".mat" extension, which are a binary data container format used in the MATLAB. The other is "CSV" for the files with ".csv" extension, in which the data are stored as a delimiter-separated text format. Apart from these, there is a file named "station_catalogue.csv". It contains the basic information and multi-year streamflow indices of every hydrological station and corresponding river reach (Table 2).

**Table 2. The fields and definitions in "station_catalogue.csv".**



| Field | Definition |
| --- | --- |
| no | station ID number |
| no_ori | station ID number in the original data source |
| river | river name |
| station | station name |
| country | country code (ISO 3166) |
| latitude | latitude (decimal degree) |
| longitude | longitude (decimal degree) |
| area | catchment size (if available, km$^2$) |
| altitude | height of gauge above sea level (m) |
| start | the start year of the time series |
| end | the end year of the time series |
| years | length of time series (years); years = end - start + 1 |
| miss | percentage of missing values in original streamflow records |
| Qmean | long-term average discharge (m$^3$/s) |
| Qmean1 | the long-term average discharge in January (m$^3$/s) |
| Qmean2 | the long-term average discharge in February (m$^3$/s) |
| Qmean3 | the long-term average discharge in March (m$^3$/s) |
| Qmean4 | the long-term average discharge in April (m$^3$/s) |
| Qmean5 | the long-term average discharge in May (m$^3$/s) |
| Qmean6 | the long-term average discharge in June (m$^3$/s) |
| Qmean7 | the long-term average discharge in July (m$^3$/s) |
| Qmean8 | the long-term average discharge in August (m$^3$/s) |
| Qmean9 | the long-term average discharge in September (m$^3$/s) |
| Qmean10 | the long-term average discharge in October (m$^3$/s) |
| Qmean11 | the long-term average discharge in November (m$^3$/s) |
| Qmean12 | the long-term average discharge in December (m$^3$/s) |
| Qmax1 | the maximum of daily streamflow (m$^3$/s) |
| Qmax3 | the maximum of consecutive 3-day streamflow average (m$^3$/s) |
| Qmax7 | the maximum of consecutive 7-day streamflow average (m$^3$/s) |
| Qmax30 | the maximum of consecutive 30-day streamflow average (m$^3$/s) |
| Qmin1 | the minimum of daily streamflow (m$^3$/s) |
| Qmin7 | the minimum of consecutive 7-day streamflow averages (m$^3$/s) |
| Qmin30 | the minimum of consecutive 30-day streamflow averages (m$^3$/s) |
| Q1st | the 1st percentile of daily streamflow (m$^3$/s) |
| Q5th | the 5th percentile of daily streamflow (m$^3$/s) |
| Q10th | the 10th percentile of daily streamflow (m$^3$/s) |
| Q25th | the 25th percentile of daily streamflow (m$^3$/s) |
| Q50th | the 50th percentile of daily streamflow (m$^3$/s) |
| Q75th | the 75th percentile of daily streamflow (m$^3$/s) |
| Q90th | the 90th percentile of daily streamflow (m$^3$/s) |
| Q95th | the 95th percentile of daily streamflow (m$^3$/s) |
| Q99th | the 99th percentile of daily streamflow (m3/s) |
| Qhigh | high flow event threshold (m$^3$/s) |
| Qlow | low flow event threshold (m$^3$/s) |
| RBFI | Richards-Baker flashiness index |
| RLD | rising limb density |
| FSS | the start month of flood season |
| HFD | half flow date (days) |
| HFI | half flow interval (days) |
| MMD | momentary maximum date (days since 1 January) |
| MC7FD | minimum consecutive 7-day flow date (days since 1 January) |

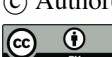



| Field | Definition |
| --- | --- |
| VY | variance of streamflow time series |
| COVY | coefficient of variation of streamflow time series |
| QCV | quartile-based coefficient of variation of streamflow time series |
| RMM | ratio of maximum to median of streamflow time series |
| GNC | Gini coefficient |
| SFDC | slope of flow duration curve |
| SDP | slope of distribution of peaks |
| VI | variability index |
| BFI | baseflow index |
| BRC | baseflow recession constant |

## 6 Conclusions and perspective

This paper presents a new global discharge indices dataset for large-sample hydrology, and is especially beneficial for hydrological analysis on streamflow regime. It includes 79 indices over 7 components of streamflow regime (*i.e.,* magnitude,

frequency, duration, changing rate, timing, variability, and recession) of 5548 river reaches globally. Before the build of indices dataset, the data collection of 15408 hydrological stations was constructed with data from GRDC, WRIS, ArcticGRO, and CHY. After that, data quality control was done on the data collection by applying an automatic detection method to identify and modify unreasonable values and to remove stations with record length less than 30 years as well as stations with the latest streamflow record date before 2000. A simple investigation into the trend and abrupt change of temporal concentration of

streamflow on a global scale was conducted as a use case of the streamflow indices dataset. Significant clusters are found in both trends and abrupt changes globally. Further investigations are needed to reveal the causes, which is beyond the scope of this study.

Compared to available similar datasets, the new indices dataset has several advantages. Firstly, it includes more indices, which could characterize streamflow regime more comprehensively. In contrast with widely used GSIM, the new indices dataset

covers indices that characterize the frequency, duration, changing rate, and recession of streamflow regime. Secondly, new river reaches and their hydrological data, especially the restricted-access hydrological data in China are included. The 33 hydrological stations cover typical river reaches of 7 largest river basins in China. Thirdly, it has longer time series which ends up no earlier than 2000. The new indices dataset has indices characterizing the frequency, duration, changing rate, and recession of streamflow regime, which are very important indices to study, for example, flow regime changes. As for the

indices of magnitude, timing, and variability, the new indices dataset includes more indices, representing more comprehensive characterizing of such streamflow components. For example, the slope of flow duration curve, slope of distribution of peaks, and variability index, which represent the variability of seasonal water balance, peak discharges, and flow duration curve respectively, are included in the new dataset. The indices time series in the new dataset are available till 2021, with lengths varying from 30 to 215 years and an average length of around 66 years. By comparison, taking GSIM as an example, the

indices time series in GSIM are between 1806 and 2016 with lengths varying from 1 to 208 years and an average length of around 38 years. Besides, more than 3500 stations' indices time series of the new dataset are available after 2017. With regard to China's hydrological data, most other datasets neither include such number of stations nor the latest records in the past two decades. For instance, all indices time series of GSIM are not available after 2004 and the time spans are from 1947 to 2004 with an average length of 12 years. Comparatively, all indices time series of this new dataset are available until 2020 and the

time spans are from 1947 to 2020 with an average length of 54 years.

This new dataset is more comprehensive and covers most common indices for streamflow regime analyses on a global scale. With the dataset, large-sample studies, such as research on streamflow regime, will become easier without spending time



collecting and handling raw streamflow records. This new dataset is a valuable source to the hydrology community to fill the gap of research on the variability, timing, changing rate, etc. of daily streamflow from a big-picture perspective. Moreover, in some cases related to hydrological risks analysis, catchment classification, hydrological model calibration and so on, the new dataset can be also useful if no original records are available.

**Author contribution**

LJ conceived the idea. LJ and XC conceptualized the study. XC, YL and LJ curated the data; XC compiled the data, performed the analyses and produced the figures; All authors contributed to the original draft of the paper.

**Competing interests**

The contact author has declared that neither they nor their co-authors have any competing interests.

**Acknowledgements**

The authors wish to express their gratitude to all the data providers, the Global Runoff Data Centre (GRDC), the India-Water Resources Information System (WRIS), and the Arctic Great Rivers Observatory (ArcticGRO) for their efforts in archiving streamflow observations and sharing the data publicly. The authors are also grateful to the developers of TOSSH toolbox for providing such a useful tool. The help from Rongrong Li from Wuhan University is highly appreciated for collecting streamflow data.

**Financial support**

This study is financially supported by the research startup grants (Y01296129; Y01296229).





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
