# Peer review of "A global streamflow indices time series dataset for large-sample hydrological analyses on streamflow regime (until 2022)"

_Earth System Science Data, 2023_

## Author Response (AR1)

**Response to Reviewer#1's Comments**

Dear Ionut Cristi Nicu,

We greatly appreciate your time and effort in reviewing our manuscript. Thanks for your approval and helpful comments. Below are our point-to-point replies.

1. *The manuscript needs to be seen by a native English speaker.*
**Response:**
We have polished the sentences of our manuscript.

2. *Abstract: as far as I know, the link to the dataset should be added here*
**Response:**
Yes, we have added it in the revised manuscript. See L26.

3. *L12-13: correct would be "…there is a lack of a comprehensive…"*
**Response:**
Thanks for your suggestion. We have corrected it. See L13.

4. *L26-27: I think soil erosion could be easily added here*
**Response:**
Yes. Soil erosion is also influenced by streamflow regime. We have added it in *Section 6 Conclusions and perspective*. See L467-473 for details.

5. *L67: "it is without doubt…" please, rephrase*
**Response:**
Thank you. We have deleted this phrase.

6. *L187: please, explain why*
**Response:**
5% is a threshold that we define as an acceptable missing ratio for indices calculation. We referred to several publications (Tramblay et al., 2021; Sauquet et al., 2021; Gudmundsson et al., 2018) and finally determined this value as a commonly acceptable value. In the revised manuscript, all records are included in the indices dataset with a quality flag. See L343-348 for details.

7. *Fig. 5a. If you could make a zoom on Europe and then add it to the figure, that would be perfect. As many data overlap, would be nice if the reader could see the whole data only for Europe*
**Response:**
Thanks for your advice. It is hard to show the whole data on the manuscript's figure. Therefore, we provide an additional vector graphic that shows all the stations clearly without overlap since the vector graphic can be zoomed in infinitely without losing any detail. The vector graphic can be accessed at https://doi.org/10.57760/sciencedb.07227.

8. *Section 6 as a whole is more than welcome and could be further developed*
*L303: a few more details could be offered here, on the direction of future studies that could use this dataset to focus on more specific hydrological issues at a local to regional scale*
**Response:**

Thanks for your kind suggestion. We have expanded this section and offered more details about relevant future studies on specific hydrological issues based on our dataset. See L462-473 for details.

**Response to Reviewer#2's Comments**

Dear Anonymous Referee,

Thanks for your helpful and insightful comments. We have carefully revised our manuscript with regard to product comparison analysis and English language. Below are our point-to-point replies.

1. *The URL and DOI of the dataset should be provided at the end of the abstract.*
**Response:**
Thanks for informing us. We have added it in the revised manuscript. See L26.

2. *Ln 67-89, many references were cited to demonstrate the importance of the indices. These references were all cited as "who did …". It is recommended to further summarize the references and enrich the expression forms.*
**Response:**
Thanks for your recommendation. We have polished these sentences to make them more concise and comprehensive with enriched sentence patterns. See L80-97 for details.

3. *Ln 159, the mean value of log(Q+0.01) and 6 times of the standard deviation can be added in Fig.2C. The examples of flood rather than outlier may be also pointed out in the example if possible. This may help readers to understand. For the assumption of 6 times of standard deviation, its applicability and possible impacts should be discussed.*
**Response:**
Thanks for your advice. This approach is originally proposed by Tank et al. (2009) for evaluating temperature series. Gudmundsson et al. (2018) introduced the $\log(Q+0.01)$ and 6 times the standard deviation and used the method to do quality control of daily values for the construction of GSIM dataset. However, the criterion primarily relies on subjective assumptions. The applicability and possible impacts of this method actually have not been discussed by anyone yet. Maybe correct values will be erroneously flagged as incorrect. Therefore, we decided not to use this disputable method. See L190-197 for details.

4. *Ln 303-320, the author briefly compared the dataset with existing data in the conclusion section. Comparative analysis is an important part of evaluating the characteristics and strengths of a dataset. It is recommended to add a separate section for comparative analysis. Quantitative analysis and examples may help to demonstrate the characteristics of the dataset.*
**Response:**
Thanks for your suggestion. We have added a separate section (*section 4 A comparative analysis*) for comparative analysis in the revised manuscript. We mapped the trends in annual mean and percentiles of streamflow during 1970 to 2022 and made a comparison with other studies' results. See L365-383 for details. Besides, in section *6 Conclusions and perspective*, we have rewritten the comparison part to highlight the advantages of our dataset. See L451-458 for details.

5. *The tense used in the paper needs to be checked and revised. For example, the tenses of Ln 184-187 are inconsistent. There are many similar problems.*
**Response:**

Thanks for pointing out these problems. We have carefully checked the tense of the whole manuscript.

**Response to Reviewer#3's Comments**

Dear Anonymous Referee,

We greatly appreciate your constructive and enlightening comments, which has helped us substantially improve our dataset. We have expanded our dataset by including stations of other available databases, *i.e.*, USGS, HYDAT, ANA, CCCRR, and BOM. Currently, the dataset covers 41263 stations. The manuscript has also been revised by adding contents about data merging and formatting, and comparative analysis. We believe the revised dataset and manuscript will be satisfying. Below are our point-to-point replies.

1. *Databases. You merged the GRDC, WRIS, ArcticGRO, and CHY databases. Why did you not use many of the other sources that are available, such as USGS, HYDAT, ARCTICNET etc.? Your selection of databases resulted in a total of 9171 stations with daily data vs. 35002 stations in GSIM. On top of that, only 5548 timeseries could be used for the calculation of indices. That is a huge difference and analyses will produce different results on a global scale.*
**Response:**
Thanks for your question. We have expanded our dataset and included as many databases as possible. Some databases are not included for various reasons. For example, A Regional Hydrographic Data Network for the Pan-Arctic Region (ARCTICNET) is not included as the data have not been updated for a long time and are outdated with the latest records at around 2001 (Lammers et al., 2016, 2001). European Flow Regimes from International Experimental and Network Data (EWA) are not incorporated since another database we have included has integrated the database. Eventually, our new dataset covers 41263 stations, which is larger than GSIM. See L115-170 for details.

2. *Databases: merging and formatting. You do not describe how you merged and formatted the timeseries of the different databases. Did you check that there were no duplicate stations across databases? How did you handle different data formats, e.g., did some data come with flags and if yes did you include them? How did you check for and merge metadata, e.g., did you check gauge locations, was catchment information included?*
**Response:**
Thanks for your questions. We have added relevant sentences to describe the merging and formatting. The duplicate stations have been identified and removed (see L128-137). Flags have been attached to every record, station, and yearly index according to specific rules. Original flags from databases have been translated into the standardized flags. (see L184-208 for details). The metadata are merged according to the fields of our dataset's fields (see L138-141 and Table 5). We did not check the gauge locations since there is no way to judge whether one station's location is right or wrong. As for the catchment information, we have included the catchment information that databases provide.

3. *Quality checks of the streamflow indices. Your quality control procedures are based on QC of the timeseries (which are the exact same filtering methods as GSIM used) and an assessment of record lengths and missing data. However, you did not perform (or did not describe in the paper) any quality checks on the indices timeseries themselves. Were there outliers within regions, and if yes can you explain them? Can you determine the reliability of the indices based*

*on the quality of the underlying daily timeseries? What about abrupt shifts in the timeseries (i.e. from rating curve updates, instrumentation changes etc.)?*

**Response:**

Thanks for your questions. We have attached quality flags to every index value according to specific criteria. The flags represent the quality of index values (see L342-348). If you mean that we did not perform homogeneity tests, we think it is not necessary. Whether there are outliers within regions or whether there are abrupt shifts in the timeseries is beyond the scope of the manuscript. There are increasing studies focused on the non-stationarity of streamflow time series and attribution. The causes of inhomogeneity or non-stationarity could be manyfold and should be investigated with a more detailed case-by-case analysis (Tramblay et al., 2021). Our dataset is a good material for these studies. In terms of shifts caused by changes of measures, corresponding correction should have been done by data providers in the phase of compilation of database as only they know the details and how to perform a correction, which is out of the scope of our work. We could guarantee that the reliability of the indices values is determined by the quality of the underlying daily records, but the quality of the underlying daily records is only determined by the providers.

4. *Indices based on baseflow estimates. The GSIM paper outlines certain issues relating to calculating indices based on baseflow, which made them decide to not include any. However, you provide recession indices without addressing any of the concerns outlined by GSIM.*

**Response:**

Thanks for your questions. We do not find relevant sentences in both Do et al. (2018) and Gudmundsson et al. (2018), but find sentences in Gudmundsson et al. (2018) as follows: "*Note also that index selection was limited to those that can be computed without a base period, which excludes many; examples include "the number of days in a year, or season, for which daily values exceed a time-of-year-dependent threshold" (Zhang et al., 2005), drought deficit volumes (Loon and Anne, 2015; Tallaksen et al., 1997) and anomalies with respect to a climatological normal (McKee et al., 1993; Shukla and Wood, 2008). There are two reasons for excluding these indices*". The term "*base period*" is not equivalent to baseflow.

5. *Overview and presentation of global indices. The paper misses a section giving an overall summary of global statistics for the indices generated. For example, a table providing the mean, max, and min of each index. Such a summary is also important as a quality check and can be used to compare the results to other studies that have provided global streamflow statistics.*

**Response:**

Thanks for your helpful advice. We have added a separate section (*section 4 A comparative analysis*) for comparative analysis on a global scale. Global trends in annual mean and percentiles of streamflow during 1970 to 2022 are mapped and compared with other studies' results (see L365-383 for details). Besides, we have provided global statistics of some indices named *"Statistics.xlsx"* in our dataset.

6. *Overall structure and presentation of the paper. Sections 3.2 and 4 are interesting but draw conclusions beyond the scope of this paper. For example, relating trends in streamflow to climatological drivers or land cover changes or other anthropogenic interference, without any robust analyses backing up these statements. Stick to a description of the dataset and a*

*presentation of the data. Further on, the text contains many repetitions, use of casual language, grammar errors, and sentence structures that do not flow well. I recommend asking an external party to review your writing.*

**Response:**

Thanks for your advice. We have replaced the Section 4 with a comparative analysis mentioned in Reply #5. As to the Sections 3.2, we have retained and polished it since it gives an intuitive impression of our indices time series. The text and sentence structures have been improved.

*7. I have accessed the .csv files only, as I do not use Matlab myself. The data is easy to access and to download, the overall description and citation information is clear, and the metadata is easy to find and well described. However, since this is a global-scale dataset which will attract researchers interested in large-scale comparative analyses, I would strongly recommend merging the 5548 separate time-series files into one csv and providing this file additional to the separate location-specific csv's. This way all information can be easily accessed using R or Python. Looking closely at a few of the individual files, they all start at the year 1806 and therefore contain much empty cells. I suggest removing the empty rows, merging the files and adding one column for station ID.*

**Response:**

Thanks for your recommendation. We have revised the dataset as you suggest. Index-specific .csv files have been created with all stations in one .csv file. The empty rows in location-specific .csv have been removed. See our dataset for details.

*8. The metadata contains information about catchment area. Does this refer to catchment area of the entire river reach or the contributing area upstream of the gauge location? Please specify (also in the paper).*

**Response:**

Catchment area refers to the contributing area upstream of the gauge location. We have specified it in the manuscript (see Table 5).

*9. I mapped the MeanQ, Qmax, and Qmin and noticed a large region north-central Canada with no values, while they do contain values for other indices. This is a little suspicious to me. How can you calculate certain indices but not mean Q?*

**Response:**

Thanks for informing us. This is due to the difference of algorithms for different multi-year indices. Some multi-year indices, for example multi-year Qmean, were calculated by taking the average of corresponding yearly Qmean, while other multi-year indices, for example multi-year Q50th, were calculated by taking the median value of the whole daily time series. When there are lots of missing data in every year, all the yearly Qmean will be set to missing value, and so will the multi-year Qmean. In contrast, for multi-year Q50th, the missing ratio has no influence on the calculation based on the whole daily time series. We have revised the algorithms to keep these indices consistent.

**References:**

Do, H. X., Gudmundsson, L., Leonard, M., and Westra, S.: The Global Streamflow Indices and Metadata Archive (GSIM) – Part 1: The production of a daily streamflow archive and metadata, Earth Syst. Sci. Data, 10, 765-785, 10.5194/essd-10-765-2018, 2018.

Gudmundsson, L., Do, H. X., Leonard, M., and Westra, S.: The Global Streamflow Indices and Metadata Archive (GSIM) – Part 2: Quality control, time-series indices and homogeneity assessment, Earth Syst. Sci. Data, 10, 787-804, 10.5194/essd-10-787-2018, 2018.

Lammers, R. B., Shiklomanov, A. I., Vörösmarty, C. J., Fekete, B. M., and Peterson, B. J.: Assessment of contemporary Arctic river runoff based on observational discharge records, Journal of Geophysical Research: Atmospheres, 106, 3321-3334, https://doi.org/10.1029/2000JD900444, 2001.

Lammers, R. B., Shiklomanov, A. I., Vörösmarty, C. J., Fekete, B. M., and Peterson, B. J.: R-ArcticNet, A Regional Hydrographic Data Network for the Pan-Arctic Region (ISO-image of CD-ROM), PANGAEA [dataset], 10.1594/PANGAEA.859422, 2016.

Sauquet, E., Shanafield, M., Hammond, J. C., Sefton, C., Leigh, C., and Datry, T.: Classification and trends in intermittent river flow regimes in Australia, northwestern Europe and USA: A global perspective, Journal of Hydrology, 597, 126170, https://doi.org/10.1016/j.jhydrol.2021.126170, 2021.

Tank, A. M. G. K., Zwiers, F. W., and Zhang, X.: Guidelines on analysis of extremes in a changing climate in support of informed decisions for adaptation, World Meteorological Organization2009.

Tramblay, Y., Rouché, N., Paturel, J. E., Mahé, G., Boyer, J. F., Amoussou, E., Bodian, A., Dacosta, H., Dakhlaoui, H., Dezetter, A., Hughes, D., Hanich, L., Peugeot, C., Tshimanga, R., and Lachassagne, P.: ADHI: the African Database of Hydrometric Indices (1950–2018), Earth Syst. Sci. Data, 13, 1547-1560, 10.5194/essd-13-1547-2021, 2021.

---

## Referee Report (RR1)

Dear Anonymous Referee,
We greatly appreciate your constructive and enlightening comments, which has helped us substantially improve our dataset. We have expanded our dataset by including stations of other available databases, i.e., USGS, HYDAT, ANA, CCCRR, and BOM. Currently, the dataset covers 41263 stations. The manuscript has also been revised by adding contents about data merging and formatting, and comparative analysis. We believe the revised dataset and manuscript will be satisfying. Below are our point-to-point replies.

**1. Databases. You merged the GRDC, WRIS, ArcticGRO, and CHY databases. Why did you not use many of the other sources that are available, such as USGS, HYDAT, ARCTICNET etc.? Your selection of databases resulted in a total of 9171 stations with daily data vs. 35002 stations in GSIM. On top of that, only 5548 timeseries could be used for the calculation of indices. That is a huge difference and analyses will produce different results on a global scale.**
Response: Thanks for your question. We have expanded our dataset and included as many databases as possible. Some databases are not included for various reasons. For example, A Regional Hydrographic Data Network for the Pan-Arctic Region (ARCTICNET) is not included as the data have not been updated for a long time and are outdated with the latest records at around 2001 (Lammers et al., 2016, 2001). European Flow Regimes from International Experimental and Network Data (EWA) are not incorporated since another database we have included has integrated the database. Eventually, our new dataset covers 41263 stations, which is larger than GSIM. See L115-170 for details.

Great!

**2. Databases: merging and formatting. You do not describe how you merged and formatted the timeseries of the different databases. Did you check that there were no duplicate stations across databases? How did you handle different data formats, e.g., did some data come with flags and if yes did you include them? How did you check for and merge metadata, e.g., did you check gauge locations, was catchment information included?**
Response: Thanks for your questions. We have added relevant sentences to describe the merging and formatting. The duplicate stations have been identified and removed (see L128-137). Flags have been attached to every record, station, and yearly index according to specific rules. Original flags from databases have been translated into the standardized flags. (see L184-208 for details). The metadata are merged according to the fields of our dataset's fields (see L138141 and Table 5). We did not check the gauge locations since there is no way to judge whether one station's location is right or wrong. As for the catchment information, we have included the catchment information that databases provide.

OK, one common error in gauge location data is too much rounding of the coordinates' decimal degree, which could be easily tracked. Other methods of checking the location would be to inspect if points fall close to a river using satellite imagery. However, I appreciate that this last step is very time-intensive and beyond the scope of this work. But you could add a sentence to indicate that this was not done.

**3. Quality checks of the streamflow indices. Your quality control procedures are based on QC of the timeseries (which are the exact same filtering methods as GSIM used) and an assessment of record lengths and missing data. However, you did not perform (or did not describe in the paper) any quality checks on the indices timeseries themselves. Were there outliers within regions, and if yes can you explain them? Can you determine the reliability of the indices based on the quality of the underlying**

**daily timeseries? What about abrupt shifts in the timeseries (i.e. from rating curve updates, instrumentation changes etc.)?**

Response: Thanks for your questions. We have attached quality flags to every index value according to specific criteria. The flags represent the quality of index values (see L342-348). If you mean that we did not perform homogeneity tests, we think it is not necessary. Whether there are outliers within regions or whether there are abrupt shifts in the timeseries is beyond the scope of the manuscript. There are increasing studies focused on the non-stationarity of streamflow time series and attribution. The causes of inhomogeneity or non-stationarity could be manyfold and should be investigated with a more detailed case-by-case analysis (Tramblay et al., 2021). Our dataset is a good material for these studies. In terms of shifts caused by changes of measures, corresponding correction should have been done by data providers in the phase of compilation of database as only they know the details and how to perform a correction, which is out of the scope of our work. We could guarantee that the reliability of the indices values is determined by the quality of the underlying daily records, but the quality of the underlying daily records is only determined by the providers.

Great that you added flags. What I meant with 'quality checks on the indices' and 'outliers' was performing simple visual checks, by mapping the indices on a global map as I did in #9. This does not have to be an extensive analysis, but you can quickly identify some unrealistic data, if present.
True, the quality of the underlying records is first the responsibility of the provider. However, when creating a dataset such as this one, there is an opportunity as automated checks on for example non-stationarity can reveal issues that individual data providers were not able to catch. I agree that this is not absolutely necessary for publication, so this is your choice.

**4. Indices based on baseflow estimates. The GSIM paper outlines certain issues relating to calculating indices based on baseflow, which made them decide to not include any. However, you provide recession indices without addressing any of the concerns outlined by GSIM.**

Response: Thanks for your questions. We do not find relevant sentences in both Do et al. (2018) and Gudmundsson et al. (2018), but find sentences in Gudmundsson et al. (2018) as follows: "Note also that index selection was limited to those that can be computed without a base period, which excludes many; examples include "the number of days in a year, or season, for which daily values exceed a time-of-year-dependent threshold" (Zhang et al., 2005), drought deficit volumes (Loon and Anne, 2015; Tallaksen et al., 1997) and anomalies with respect to a climatological normal (McKee et al., 1993; Shukla and Wood, 2008). There are two reasons for excluding these indices". The term "base period" is not equivalent to baseflow.

You are right. My apologies for this oversight.

**5. Overview and presentation of global indices. The paper misses a section giving an overall summary of global statistics for the indices generated. For example, a table providing the mean, max, and min of each index. Such a summary is also important as a quality check and can be used to compare the results to other studies that have provided global streamflow statistics.**

Response: Thanks for your helpful advice. We have added a separate section (section 4 A comparative analysis) for comparative analysis on a global scale. Global trends in annual mean and percentiles of streamflow during 1970 to 2022 are mapped and compared with other studies' results (see L365-383 for details). Besides, we have provided global statistics of some indices named "Statistics.xlsx" in our dataset.

**6. Overall structure and presentation of the paper. Sections 3.2 and 4 are interesting but draw conclusions beyond the scope of this paper. For example, relating trends in streamflow to climatological drivers or land cover changes or other anthropogenic interference, without any robust analyses backing up these statements. Stick to a description of the dataset and a presentation of the data. Further on, the text contains many repetitions, use of casual language, grammar errors, and sentence structures that do not flow well. I recommend asking an external party to review your writing.**

Response: Thanks for your advice. We have replaced the Section 4 with a comparative analysis mentioned in Reply #5. As to the Sections 3.2, we have retained and polished it since it gives an intuitive impression of our indices time series. The text and sentence structures have been improved.

**7. I have accessed the .csv files only, as I do not use Matlab myself. The data is easy to access and to download, the overall description and citation information is clear, and the metadata is easy to find and well described. However, since this is a global-scale dataset which will attract researchers interested in large-scale comparative analyses, I would strongly recommend merging the 5548 separate time-series files into one csv and providing this file additional to the separate location-specific csv's. This way all information can be easily accessed using R or Python. Looking closely at a few of the individual files, they all start at the year 1806 and therefore contain much empty cells. I suggest removing the empty rows, merging the files and adding one column for station ID.**

Response: Thanks for your recommendation. We have revised the dataset as you suggest. Indexspecific .csv files have been created with all stations in one .csv file. The empty rows in location-specific .csv have been removed. See our dataset for details.

**8. The metadata contains information about catchment area. Does this refer to catchment area of the entire river reach or the contributing area upstream of the gauge location? Please specify (also in the paper).**

Response: Catchment area refers to the contributing area upstream of the gauge location. We have specified it in the manuscript (see Table 5).

**9. I mapped the MeanQ, Qmax, and Qmin and noticed a large region north-central Canada with no values, while they do contain values for other indices. This is a little suspicious to me. How can you calculate certain indices but not mean Q?**

Response: Thanks for informing us. This is due to the difference of algorithms for different multi-year indices. Some multi-year indices, for example multi-year Qmean, were calculated by taking the average of corresponding yearly Qmean, while other multi-year indices, for example multiyear Q50th, were calculated by taking the median value of the whole daily time series. When there are lots of missing data in every year, all the yearly Qmean will be set to missing value, and so will the multi-year Qmean. In contrast, for multi-year Q50th, the missing ratio has no influence on the calculation based on the whole daily time series. We have revised the algorithms to keep these indices consistent.

OK. As mentioned also in #3, performing these kind of checks (i.e., visualizing on a global map) is not a lot of work, so make sure that you have done them also on other indices. In my opinion, it is not acceptable to have these relatively easily solvable algorithm errors in a published dataset.